# Adaptation Properties Allow Identification of Optimized Neural Codes

**Luke Rast**
Harvard University
lukerast@g.harvard.edu

**Jan Drugowitsch**
Department of Neurobiology
Harvard Medical School
jan_drugowitsch@hms.harvard.edu

## Abstract

The adaptation of neural codes to the statistics of their environment is well captured by efficient coding approaches. Here we solve an inverse problem: characterizing the objective and constraint functions that efficient codes appear to be optimal for, on the basis of how they adapt to different stimulus distributions. We formulate a general efficient coding problem, with flexible objective and constraint functions and minimal parametric assumptions. Solving special cases of this model, we provide solutions to broad classes of Fisher information-based efficient coding problems, generalizing a wide range of previous results. We show that different objective function types impose qualitatively different adaptation behaviors, while constraints enforce characteristic deviations from classic efficient coding signatures. Despite interaction between these effects, clear signatures emerge for both unconstrained optimization problems and information-maximizing objective functions. Asking for a fixed-point of the neural code adaptation, we find an objective-independent characterization of constraints on the neural code. We use this result to propose an experimental paradigm that can characterize both the objective and constraint functions that an observed code appears to be optimized for.

## 1 Introduction

It is well known that neural codes adapt to the distribution of stimulus inputs that they receive on a variety of timescales. For example, in early fly vision, in addition to being well adapted to unchanging statistics of natural images [1], neural sensitivity adapts to both fast- and medium- timescale changes in the distribution of stimuli [2–4]. Similar effects have been observed in mammalian vision [5, 6]. Such adaptation effects are often viewed through the lens of the efficient coding hypothesis [7, 8], which states that neural codes are well-adapted to their task and environmental contexts, and so can be modeled as solutions to task and environment dependent optimization problems. This is a reasonable modeling approach since we expect the result of an adaptation process to be stable in the sense of a first-order optimality condition: there are no directions that the code will continue to adapt along once it has come to equilibrium.

Several recent works in efficient coding have focused on optimizing the local Fisher information in the neural code. Starting from mutual information approximating objective functions and independent Poisson neural populations [9], this approach has been generalized to accommodate a variety of objective functions [10–13] and metabolic constraints [10, 11, 14] under various parametrized neural activity models. In an activity distribution agnostic formulation, these Fisher efficient codes have also been shown to predict general behavioral effects for various objective functions [15–18]. This diversity of result leads to hope that we can fully characterize the influence of objectives, constraints, and activity noise distributions in Fisher efficient codes and, in so doing, describe observed

neural codes in terms of the optimization problem that they appear to be solving. Similar recent approaches have fit objective functions to observed behavior [19, 20] and neural activity [21] by inverse reinforcement learning/optimal control, although this often leads to underspecified problems.

In this work we show that, for Fisher information based efficient codes with one-dimensional stimulus variables, the objective and constraint functions that a code is optimized for can be fully characterized by examining how the neural code adapts to changes in the stimulus distribution. Working with flexible objective functions and constraints in a minimally parameterized setting, we first provide solutions to broad classes of Fisher efficient coding problems, generalizing previous solutions. We show that constraints enforce deviation from classical histogram equalization behaviors, while changing the objective results in qualitative changes to the adaptation behavior of the neural code. These solutions allow us to derive tests for the presence of constraints and for mutual information approximating objectives. Next, asking for a stationary-point of the neural code adaptation, we find an objective independent characterization of constraints on the neural code. Using this result, we propose an experimental paradigm to characterize both objective functions and constraints based on the observed adaptation properties of neural codes.

## 2   Results

Let $s$ be a one-dimensional stimulus of interest, drawn from a distribution, $p_s(s)$ as specified by a task or environment. This stimulus is encoded in the activity $r$ of one or more neurons in a population according to an encoding model $p(r|s)$. In an efficient code, this encoding has been optimized for performance under a specific objective while obeying particular constraints imposed on neural activity. Specification of an efficient coding problem thus requires choosing [22]: (i) what features of the code can be optimized, (ii) how the performance of the code is measured, and (iii) what constraints are imposed on neural activity. Here, we solve in general a class of efficient coding problems with: optimized sufficient statistics, Fisher information-based objective functions, and activity-dependent constraints.

### 2.1   Approach

Our approach makes three key assumptions. First, we assume that the one-dimensional stimulus of interest is coded noisily around a one-dimensional submanifold of neural activity. The submanifold itself is held fixed while we optimize how different stimuli are associated with different locations in this neural submanifold. This assumption is made implicitly in a wide range of previous work [9–18]. Formally, take $\theta(s)$ to be a *sufficient statistic* for the stimulus in the neural activity,

$$p(\boldsymbol{r}|s) = p(\boldsymbol{r}|\theta(s)), \tag{1}$$

so that $\theta(s)$ captures everything about stimulus $s$ that is encoded in neural activity $r$. We assume that this $\theta(s)$ is *continuous* and *strictly increasing* and that $\theta$ spans a *finite interval*. This deterministic mapping $\theta(s)$ will be optimized, while the corresponding neural activity distribution $p(r|\theta)$ is fixed but left unspecified, so that the results apply to any such encoding. For example, for a single neuron with Poisson activity, we could choose $p(r|\theta) = \text{Pois}\,(r|\theta)$ in order to optimize the neuron's (monotonic) tuning curve, $\theta(s)$ (Fig. 1a). Alternatively, in a neural population with bell-shaped turning curves, $p(\boldsymbol{r}|\theta)$ could specify the relative mean activities across different neurons, as well as their individual noise, while $\theta(s)$ determines how the neurons cover the stimulus space (Figs. 1b & c). Irrespective of the encoding model's details, the $s \to \theta \to r$ separation splits it into two steps. The fixed $\theta \to r$ relationship parameterizes the neural activity with a one-dimensional parameter, $\theta$, while the optimized $s \to \theta$ relationship specifies how much of stimulus space is associated with each interval of parameter space (e.g., Fig. 1c). Our results apply to any encoding model that splits in this way.

Our second and third assumptions concern the form of the objective and constraint functions that the neural code is optimized for. We assume that the objective depends on the *Fisher information*, or local sensitivity of the code, while constraints depend on the level of *neural activity*. Formally, the code optimizes

$$\max \; \mathbb{E}_{p_s(s)} \left[ f\left( \sqrt{I_s(s)} \right) \right] \quad \text{s.t.} \quad \mathbb{E}_{p_s(s)p(\boldsymbol{r}|s)} \left[ h(\boldsymbol{r}) \right] \leq M, \tag{2}$$

where $\mathbb{E}_{p(\cdot)}$ is the expectation with respect to $p(\cdot)$. That is, both sensitivity and activity are averaged across stimuli, while activity is additionally averaged over neural noise. The Fisher information,

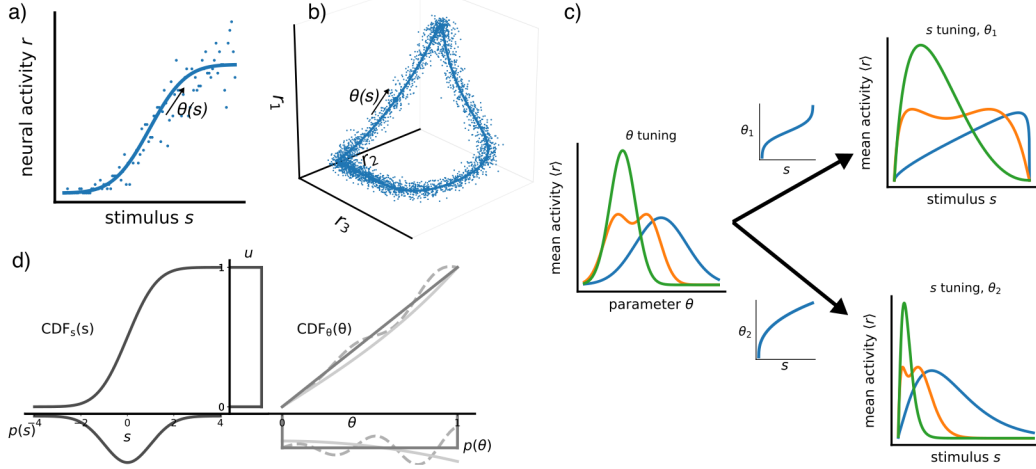

Figure 1: Our efficient coding setup. (a) & (b) Example encoding models, $p(\boldsymbol{r}|\theta)$, for (a) single neuron and (b) neural population. In (a), $\theta$ specifies the mean activity (tuning) of neurons, while noise is be assumed Poisson. In (b), the encoding model specifies a particular one-dimensional submanifold of neural activity (blue line) that the stimulus is coded along. (c) Effect of different $\theta(s)$ functions on the tuning of neurons in example (b). Functions $\theta(s)$ determine the "speed" that we move along the one-dimensional manifold, but not the relative average activity among different neurons (colors). (d) Due to monotonicity, the $\theta(s)$ mapping can be characterized in terms of cumulative distributions, $\theta(s) = \mathrm{CDF}_\theta^{-1}(\mathrm{CDF}_s(s))$, or probability densities $\mathrm{d}\theta/\mathrm{d}s = p_s(s)/p_\theta(\theta)$, illustrated here are different $\theta(s)$ mappings (line style) for the same $p_s(s)$.

defined as $I_s(s) = \mathbb{E}_{p(\boldsymbol{r}|s)}\left[\left(\frac{\partial}{\partial s}\log p\left(\boldsymbol{r}|s\right)\right)^2\right]$, measures how well neural activity $\boldsymbol{r}$ allows us to discriminate stimulus values close to $s$ [23, 24]. In Eq. (2), both the local sensitivity and the local activity are wrapped in functions, $f(\sqrt{I_s})$ and $h(\boldsymbol{r})$, which specify respectively how good or bad additional information or activity are. We assume that $f(\cdot)$ is *increasing* (i.e. more information is better), and $h(\cdot)$ is *non-negative* (i.e. constrained resources can be used up, but not produced by neural activity). $f(\cdot)$ is taken to be a function of the square root of Fisher information (rather than Fisher information itself) for notational convenience: we show in the supplementary material that $f(\cdot)$ must be *concave* and sub-linear at infinity to have well-behaved solutions. As long as the objective and constraint functions satisfy these properties and are fixed when optimizing the neural code, they can otherwise be arbitrary (continuous). We will treat $h(\cdot)$ as one-dimensional, for example, $h(\boldsymbol{r}) = \mathbf{1}^T\boldsymbol{r}$, which puts an upper bound on the average total activity of the neurons. However, the results can easily be extended to higher-dimensional $h$ (i.e. multiple constraints) (see supplementary material).

Combining our assumptions, the overall problem becomes the following optimization of $\theta(s)$:

$$\max_{\theta(s)} \mathbb{E}_{p_s(s)}\left[f\left(\sqrt{I_\theta(\theta)}\frac{\mathrm{d}\theta}{\mathrm{d}s}\right)\right] \quad \text{s.t.} \quad \mathbb{E}_{p_s(s)}\left[C(\theta)\right] \leq M, \quad \frac{\mathrm{d}\theta}{\mathrm{d}s} > 0, \quad \int \frac{\mathrm{d}\theta}{\mathrm{d}s}\mathrm{d}s = L_\theta. \quad (3)$$

The last two constraints express our assumptions on $\theta(s)$ in the encoding model. In this formulation, neural activity dependence has been re-expressed in terms of the activity parameter through $p(\boldsymbol{r}|\theta)$. This encoding model enters the optimization both through $I_\theta(\theta)$, which expresses the intrinsic sensitivity of neural activity $\boldsymbol{r}$ to changes in parameter $\theta$, and through $C(\theta)$, which denotes the average level of constraint usage for parameter $\theta$. These functions inherit their properties from $p(\boldsymbol{r}|s)$ and $h(\boldsymbol{r})$ (in the case of $C(\theta)$) and thus are continuous and positive, but otherwise left fixed but arbitrary. This formulation provides an appealing level of generality as it reveals principles of efficient codes irrespective of their specific neural instantiation. However, it also means that the results do not make direct predictions about neural activity, since this additionally requires specifying or identifying an encoding model $p(\boldsymbol{r}|\theta)$. Overall, we aim to find the optimal parameter mapping $\theta(s)$ that solves Eq. (3). This variational optimization problem can be solved by Euler-Lagrange or the Pontryagin Maximality Principle in standard fashion [25, 26]. A central goal of this work will be to ask what

effect the choices of $p_s(\cdot)$, $I_\theta(\cdot)$, $f(\cdot)$ and $C(\cdot)$ have on the optimal solution, under forgiving assumptions about their forms. See supplementary material for a full list of assumptions and for the solution method. We start with analytic solutions for relevant special cases, below.

## 2.2 Special-case analytic solutions

Here we present three special-case solutions that generalize previously studied Fisher efficient coding problems [9–18]. Solutions to our efficient coding problem are functions $\theta(s)$, mapping stimuli to corresponding neural activity parameters. As $\theta(s)$ is continuous and monotonic, this mapping can also be characterized in terms of probability densities: sampling across input stimuli, $p_s(s)\mathrm{d}s = p_\theta(\theta)\mathrm{d}\theta$, or in terms of cumulative distribution functions: $\theta(s) = \mathrm{CDF}_\theta^{-1}(\mathrm{CDF}_s(s))$ (Fig. 1d). The cumulative distribution functions of $s$ and $\theta$ cannot always be expressed in closed form, so we characterize $\theta(s)$ by its probability density function, $p_\theta(\theta)$, from which we can recover $\theta(s)$ by (if necessary, numerical) integration. As $\theta$ determines neural activity through $p(\boldsymbol{r}|\theta)$, the $p_\theta(\theta)$ distribution tells us the weight that the efficient coding solution assigns to each part of the neural activity space; $\theta$'s with high $p_\theta(\theta)$ imply that the associated $\boldsymbol{r}$'s are more likely to occur in an optimized code, and thus preferred.

### 2.2.1 Log-objective function, $f(\sqrt{I}) \propto \log I$

If $f(\sqrt{I}) \propto \log I$, then the objective approximates mutual information in high signal-to-noise settings [9, 27]. This objective function has been well studied in a variety of encoding models [9, 11, 15, 17], notably [14]. In our formulation, these solutions can be generalized to the optimal coding solution (Fig. 2a), characterized by:

$$p_\theta(\theta) = Z^{-1}\sqrt{I_\theta(\theta)}\exp(-\lambda C(\theta)), \qquad (4)$$

an exponential family distribution with normalization factor $Z$, base measure $\sqrt{I_\theta(\theta)}$, parameter $\lambda$, and sufficient statistic $C(\theta)$. In relation to the efficient coding problem, Eq. (3), the $M$-dependent Lagrange multiplier $\lambda > 0$ is in charge of enforcing the constraint on $C(\theta)$, while the normalization, $Z$, ensures that the $\theta$ interval has the desired length $L_\theta$. In the absence of constraints (e.g., if $M \to \infty$), the optimal solution is determined by only the base measure, $p_\theta(\theta) \propto \sqrt{I_\theta(\theta)}$. Otherwise, $p_\theta(\theta)$ will be as close as possible (in the Kullback-Leiber sense) to this unconstrained solution while still satisfying the constraint. Either way, the optimal solution results in the Fisher information in neural activity $\boldsymbol{r}$ about stimulus $s$ given by

$$I_s(s) \propto p_s(s)^2 \exp(2\lambda C(\theta(s))), \qquad (5)$$

which is larger for more likely stimuli and for stronger constraints. We will return the properties of the found solution in Sections 2.4.1 and 2.4.2.

### 2.2.2 Unconstrained optimization, $M \to \infty$

Next, we generalize the power-law objective function analyses of [10–13], to examine the effect of the objective function in the absence of constraints. For our family of well-behaved objective functions, the optimal unconstrained solution (Fig. 2b) is given by

$$p_\theta(\theta) = \sqrt{I_\theta(\theta)}\frac{p_s(s(\theta))}{f'^{-1}\left(\frac{Z}{p_s(s(\theta))}\right)}. \qquad (6)$$

Here $f'^{-1}(\cdot)$ will exist due to concavity of $f$, while $Z$ remains the length-enforcing normalization factor. This solution is implicit due to its dependence on $s(\theta)$. It nonetheless shows that $p_\theta(\theta)$ follows a warped version of stimulus distribution, $p_s(s)$, where the warping is determined by $f'^{-1}(\cdot)$. Different warpings can result in substantially different adaptation behaviors of the neural code to changing stimulus distribution. For example, for power-law objective function: $f(\sqrt{I}) \sim \alpha I^\alpha$ (which agree with previous solutions [12, 13]), $\alpha \downarrow -\infty$ leads to $p_\theta(\theta) \to \sqrt{I_\theta(\theta)}p(s)$ (a $\sqrt{I_\theta}$-scaled copy of stimulus distribution) while $\alpha \uparrow 1$ (at which point the solution becomes ill behaved) gives $p_\theta(\theta) \to \sqrt{I_\theta(\theta)}p(s)^{-\infty}$ (strongly repelled from encoding likely stimuli). The log case occurs between these limits at $\alpha = 0$: $p_\theta(\theta) \propto \sqrt{I_\theta(\theta)}p(s)^0$ recovers the non-adapting parameter distribution of Eq. (4) (when unconstrained). Irrespective of the specific choice for objective function, the

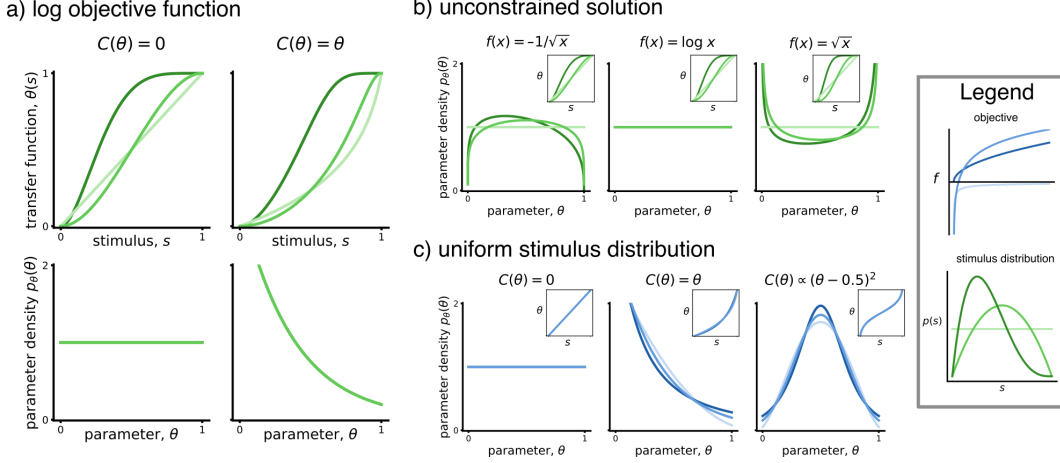

Figure 2: Special case efficient coding solutions, $p_\theta(\theta)$. The corresponding $\theta(s)$ curves are shown in the top row of (a), and insets in (b) and (c). (a) log-objective function solution under multiple constraints (columns) and stimulus distributions (colors). (b) unconstrained solution under multiple objective functions (columns) and stimulus distributions (colors). (c) uniform stimulus distribution solution under multiple constraints (columns) and objective functions (colors). Plotted in the legend, the objective function can be $-1/\sqrt{x}$ (light) $\log x$ (medium), $\sqrt{x}$ (dark), and the stimulus pdf: $\text{Beta}(1,1)$ (light), $\text{Beta}(2,2)$ (medium), $\text{Beta}(2,5)$ (dark). Constraints are either $C(\theta) = \theta$ or $C(\theta) = 7(\theta - 0.5)^2$. For constrained panels, $M = 0.3$. For all plots, $I_\theta(\theta) = 1$.

stimulus Fisher information of the unconstrained optimal solution satisfies

$$f'(\sqrt{I_s(s)}) \propto \frac{1}{p_s(s)}. \tag{7}$$

We will discuss further properties of this solution in Sec. 2.4.2.

### 2.2.3 Uniform probability density, $p_s(s) \propto 1$

Setting the stimulus distribution to be uniform allows us to find an analytical solution that illustrates how objective and constraint functions interact. Solving the optimization while assuming $p_s(s) = p_s$ yields (Fig. 2c),

$$p_\theta(\theta) = \sqrt{I_\theta(\theta)} \frac{p_s}{\hat{f}^{-1}(\lambda C(\theta) + Z)}. \tag{8}$$

Here, $\lambda$ and $Z$ are again the Lagrange multipliers in charge of enforcing constraint and normalization respectively, and $\hat{f}(\cdot)$ is the Legendre transform of $f(\cdot)$, defined as $\hat{f}(x) = f'(x)x - f(x)$, which exists uniquely because $f(\cdot)$ is concave. As for the log-objective function case, $p_\theta(\theta)$ is as close to $\sqrt{I_\theta(\theta)}$ as possible while satisfying the constraint. However, here the constraint is enforced through the inverse Legendre transform of the objective function. Equivalently, closeness is measured by a more general f-divergence [28], which, in our case, is specified by the objective function: the $-f(\cdot)$ divergence. Setting $f(x) = \log(x)$ recovers the KL divergence, and because $\hat{f}^{-1}(x) = \exp(-x - 1)$, it also recovers the solution for the log-objective function, Eq. (4). Irrespective of the specific choice of object function, the stimulus Fisher information of the optimal solution satisfies

$$I_s(s) \propto \left( \hat{f}^{-1}(\lambda C(\theta(s)) + Z) \right)^2. \tag{9}$$

This solution is discussed further in Sec. 2.4.1.

### 2.3 Reparameterization to intrinsic flat-Fisher parameters

None of the above solutions required us to specify the parameter-conditioned neural activity distribution, $p(\mathbf{r}|\theta)$, which comes into play in two places: it determines how we interpret parameter constraints in terms of activity, $C(\theta) = \mathbb{E}_{p(\mathbf{r}|\theta)}[h(\mathbf{r})]$, and it determines the amount of information

$I_\theta(\theta)$ that neural activity provides about $\theta$. In all of our special-case solutions, $\sqrt{I_\theta(\theta)}$ scales the distribution of parameters (Eqs. (4), (6) & (8)), but does not impact the stimulus Fisher information (Eqs. (5), (7) & (9)). In other words, the emphasis on different parameter values $\theta$ is re-weighted in proportion to how intrinsically informative the associated neural responses are. This is a general feature: $\sqrt{I_\theta(\theta)}$ acts as the local measure of the parameter space, as could be expected from information geometry [29, 30]. Assuming that $I_\theta(\theta)$ is strictly positive (rather than only non-negative), we can introduce a reparameterization $\theta \to \hat{\theta}$ satisfying $d\hat{\theta} = \sqrt{I_\theta(\theta)}d\theta$. This results in constant Fisher information, $I_{\hat{\theta}}(\hat{\theta}) \propto 1$, so we refer to $\hat{\theta}$ as the *flat-Fisher space*.

This reparametrization does not affect the problem that we are solving or any features of the neural response. After all, the parameter $\theta$ is inherited from our parameterization of the encoding model $p(\boldsymbol{r}|\theta)$, but is not unique: any invertible function of the parameter can provide a new parameterization of the problem, with its own parameter Fisher information and corresponding optimal parameter distribution. However, by reparameterizing to the flat-Fisher parameter space, we remove the dependence on parameter Fisher information, $I_\theta(\theta)$, from the previously derived solutions (Eqs. (4), (6) & (8)), giving $p_{\hat{\theta}}(\hat{\theta}) \propto p_\theta(\theta(\hat{\theta}))/\sqrt{I_\theta(\theta)}$. More generally, in the formulation of the optimization problem, Eq. (3), the flat space reparameterization removes explicit $I_\theta$ dependence from the objective and renders the parameter length constraint $L_\theta$ identical to a constraint on the total available stimulus Fisher information, as previously employed by [15–17] (see supplementary material). Thus, by performing the reparameterization, we have removed features of the solution that result from our initial choice of parameter. In a geometric sense, where $\theta$ parameterizes a curve through neural activity space, $\hat{\theta}$ is the *arc-length* parameterization of this curve: it tells us where we are on the curve as a function of how far (in the Fisher metric) we have already traveled. The results in this flat-Fisher parameterization are intrinsic to the neural code: Solving in terms of the stimulus Fisher information we find

$$p_{\hat{\theta}}\left(\hat{\theta}(s)\right) \propto p_s(s)/\sqrt{I_s(s)}. \tag{10}$$

Thus, the flat-Fisher space parameter distribution can be expressed in terms of measurable quantities: the stimulus Fisher information (discrimination performance) and stimulus distribution, and so can in principle be measured. In the following sections, we will work in this flat-Fisher space by default, deriving tests that can be performed by measuring the flat space parameter distribution $p_{\hat{\theta}}(\hat{\theta})$, and showing how to fit it from simulated neural data.

## 2.4 Signatures of objective and constraint functions in efficient codes

Using the special case solutions expressed in the flat-Fisher space, we can derive tests for (i) whether the neural code is shaped by constraints on neural activity, and (ii) whether the neural code optimizes a log-objective function, as well as (iii) an experimental paradigm to fit both objective and constraint functions based on how the neural code adapts to changes in the stimulus distribution. These tests will rely on the ability to probe the code using stimuli drawn from specific stimulus distributions. In order to do this we must make additional assumptions, namely that the neural code *adapts rapidly* to changes in the stimulus distribution $p_s(s)$ (as some neural codes have previously been shown to [4, 5]) and does so by optimizing the stimulus-to-parameter mapping $\theta(s)$, while the neural activity distribution $p(\boldsymbol{r}|\theta)$ is *fixed* within the timescale of the experiments. That is to say, we assume experimental control over $p_s(s)$, and stimulus-distribution-invariant neural encoding $p(\boldsymbol{r}|\theta)$, such that only $\theta(s)$ changes when adapting to different stimulus distributions.

### 2.4.1 Activity constraints enforce deviation from flat-Fisher space equalization

Our first test leverages deviations from a classic efficient coding signature: *histogram equalization* [1], which refers to the observation that some measured parameter of the neural activity (often the mean firing rate) is uniformly distributed when sampling across stimuli. Here we focus on histogram equalization behavior for the flat-Fisher parameter. Specifically, we say that 'flat-Fisher space histogram equalization' occurs when the $\hat{\theta}$ distribution is uniform:

$$p_{\hat{\theta}}(\hat{\theta}) \propto 1. \tag{11}$$

Expressed in terms of the original parameter, this distribution has a familiar form, $p_\theta(\theta) \propto \sqrt{I_\theta(\theta)}$. This is the optimal parameter distribution for both log-objective functions (Eq. (4)) and uniform stimulus distributions (Eq. (8)) when there are no constraints on activity (i.e. $\lambda = 0$). Focusing

on the uniform stimulus distribution case, we see that, regardless of the objective function that is being optimized, if the code is adapted to a uniform distribution of stimuli and there are no activity constraints, then histogram equalization will be achieved in the flat-Fisher space. Vice versa, if neural activity constraints are binding, then the efficient coding solution will deviate from such flat space equalization when adapted to a uniform stimulus distribution. The solutions are as close to flat space histogram equalization as possible while still satisfying the imposed constraint, with closeness measured by the objective function dependent $-f$-divergence. This constraint enforced deviation from histogram equalization has been noted in previous works with log objective functions [11, 14] when Fisher information was assumed to be constant. Here we see that, for any neural code that satisfies our general assumptions, under any objective function, when the code is adapted to a uniform stimulus distribution, it will deviate from flat-Fisher equalization if and only if there are constraints present on the neural activity. This property allows us to test whether such constraints are present: by allowing the neural code to adapt to $p_s(s) \propto 1$ and measuring $p_{\hat{\theta}}(\hat{\theta})$ we can test whether the code is constrained ($p_{\hat{\theta}}(\hat{\theta}) \not\propto 1$ ) or unconstrained ($p_{\hat{\theta}}(\hat{\theta}) \propto 1$).

This test can also be phrased in terms of the Fisher information in the code about the stimulus, $I_s(s)$. Eqs. (5) and (9) reveal that, when unconstrained, $I_s(s) \propto p_s(s)^2$, so that there is more information about more likely stimuli. This is intuitive because the histogram equalization solution spans a wider dynamic range of parameters across more likely stimuli. In fact, $p_{\hat{\theta}}(\hat{\theta}) \propto p_s(s)/\sqrt{I_s(s)}$ (Eq. (10)) shows that in this case, the flat-Fisher space parameter density measures the local deviation from histogram equalization.

### 2.4.2   Non-log objective functions result in stimulus distribution dependence

Our second test focuses on the *adaptation* of the neural code, which is to say, the differences in the neural code when it is optimized for different stimulus distributions, $p_s(s)$. As we saw in the unconstrained solution (Sec. 2.2.2 and Fig. 2b), a major impact of the objective function is that it changes how the optimal solution adapts to different stimulus distributions, with each objective function having its own characteristic adaptation behavior. The log objective function is unique from an adaptation perspective because its optimal parameter distribution $p_{\hat{\theta}}(\hat{\theta})$ is invariant to changes in the stimulus distribution. This happens because the optimal stimulus-to-parameter mapping $\hat{\theta}(s)$ adapts to different stimulus distributions in a way that compensates for these differences, leaving $p_{\hat{\theta}}$ unchanged (see Fig. 2a). In this unconstrained case, the log objective function will always achieve flat-Fisher histogram equalization, while non-log objective functions will not be histogram equalizing except for uniform stimulus distributions. This can be seen by expressing the solution as:

$$f'\left(\frac{p_s(s)}{p_{\hat{\theta}}(\hat{\theta}(s))}\right) p_s(s) \propto 1. \tag{12}$$

This shows that the $p_s(s)$ dependency only cancels if $f'(x) \propto 1/x$, which requires a logarithmic objective function. As we show in the supplementary material (Sec. 4.1), this property extends to general constraints: the logarithm is the only objective function whose parameter distributions are invariant to changes in the stimulus distribution, regardless of constraints. This is essentially because the stimulus distribution still enters these (differential) equations through a term of the same form as above, $f'\left(p_s(s)/p_{\hat{\theta}}(\hat{\theta})\right) p_s(s)$. Thus, neural code adaptation allows us to test whether or not the objective function is log: regardless of the presence or absence of constraints on neural activity, if changing the stimulus distribution does not impact the parameter distribution $p_{\hat{\theta}}(\hat{\theta})$ of the optimized code, then the objective function is $f(\sqrt{I}) \propto \log I$; otherwise it is not.

The log objective function is important because, under the right conditions, it approximates the mutual information [9, 27], and so should be a reasonable objective for coding in early sensory areas. In fact, when adapted to Gaussian distributions of visual angular velocity stimuli with different variances, the stimulus-to-neural firing rate relationship of H1 neurons in the blowfly visual system only differs in the scaling the of stimulus [2–4]. In other words, the activity distributions are preserved. By our log objective function test, this is evidence for a logarithmic (i.e. mutual information maximizing) objective function.

### 2.4.3 A closed-loop experimental paradigm to identify both objective and constraint functions

The tests derived thus far support detecting the presence of constraints and the presence of log-objective functions, but do not allow us to fully characterize the objective or constraint function that a code is optimized for. These previous tests suggest that the constraint impacts deviations from histogram equalization, while the objective function impacts adaptation of the neural code to different stimulus distributions. In order to disentangle their impacts on the optimized code, we first 'remove' the effects of adaptation by finding a fixed point of the neural adaptation. At this fixed point, which we will introduce in detail below, the optimal coding solution becomes independent of the objective function, and thus allows us to measure the constraint function in isolation. Once this is achieved, we can measure adaptation to another non-fixed point stimulus distribution, which depends on both constraint and objective function, and allows us to determine the latter.

Let us first describe what we mean by a *fixed point* of the neural code adaptation. Because we are examining how the neural code adapts to different stimulus distributions, we can think of this adaptation as a mapping from distributions of stimuli to distributions of neural activity parameters: $p_s(s) \to p_{\hat{\theta}}(\hat{\theta}(s))$. A fixed point of such a mapping is a stimulus distribution, $p_s(s)$, such that after adaptation to $p_s(s)$, the neural parameter distribution is proportional to this same distribution: $p_{\hat{\theta}}(\hat{\theta}(s)) \propto p_s(s)$. This fixed point can equivalently be characterized by $\hat{\theta}(s) = as + b$: a change in stimulus induces the same change in activity parameters regardless of the starting stimulus, or via Eq. (10) by $I_s(s) \propto 1$: at the fixed point, the neural code contains the same Fisher information about all stimuli. We can find the fixed point in closed form (see also supplementary material) using the proportionality of stimulus and parameter probability densities. Note that in Eq. (3), the argument to the object function equals the ratio of probability densities $p_s(s)/p_{\hat{\theta}}(\hat{\theta}(s))$, which is constant at the fixed point. This property extends to the solutions of the optimization, rendering the fixed point solution independent of the objective function. The fixed point density is given by

$$ p_{\hat{\theta}}(\hat{\theta}(s)) \propto p_s(s) \propto \exp\left(-\lambda C_{\hat{\theta}}\left(\hat{\theta}(s)\right)\right), \tag{13} $$

where $C_{\hat{\theta}}(\hat{\theta}) = C(\theta(\hat{\theta}))$ is the constraint function expressed as a function of the flat-Fisher parameter. At this stimulus distribution, the distribution of neural activity parameters is equal to the stimulus distribution and both are equivalent to the log-objective function solution (Eq. (4) expressed in flat-Fisher parameters). This intuitively makes sense because the log-objective function solution is independent of the stimulus distribution, such that it must always occupy the fixed point.

Crucially, the fixed point solution is the same regardless of the objective function that the code is optimizing. Thus, finding the fixed point gives us an objective-independent characterization of the constraint, $C_{\hat{\theta}}(\hat{\theta})$. With the constraint function in hand, we can then probe the neural code at another stimulus distribution to find the objective function. For example, with a uniform stimulus distribution, Eq. (8), if we know $C_{\hat{\theta}}(\cdot)$ and $p_{\hat{\theta}}(\cdot)$, then $\hat{f}^{-1}(\cdot)$ can be found by regression.

The only remaining piece in the puzzle is to find the fixed point stimulus distribution. This can be accomplished with a *closed-loop experimental paradigm*, which is in essence a fixed point iteration: the neural activity measured in response to stimuli drawn from one stimulus distribution determines the stimulus distribution that will be presented next. Specifically, we allow the neural code to adapt to an initially uniform stimulus distribution, $p_s(s) \propto 1$, and then measure neural responses $r_i$ to stimuli $s_i \sim p_s$. From these responses, we fit the flat-Fisher space distribution, $p_{\hat{\theta}}$, which then becomes the new stimulus distribution. The process of re-adaptation, measurement, and updating the stimulus distribution repeats until there are no changes to the stimulus distribution. At this point, $p_{\hat{\theta}} \propto p_s$, and the adaptation fixed point is achieved. The constraint can then be fit from the fixed point at the last step of this iteration and, given this constraint, the objective can be fit to the results from the first step, which had a uniform stimulus distribution. There is one edge case: if the code is unconstrained, then we know from the constraint test that the flat-Fisher parameter distribution will be uniform when the stimulus distribution is uniform. As a consequence, the iteration will terminate on the first step. If this is the case, another stimulus distribution is needed to identify the objective function. Because we now know that the code is unconstrained, this can be accomplished using any stimulus distribution and the general unconstrained solution, Eq. (6).

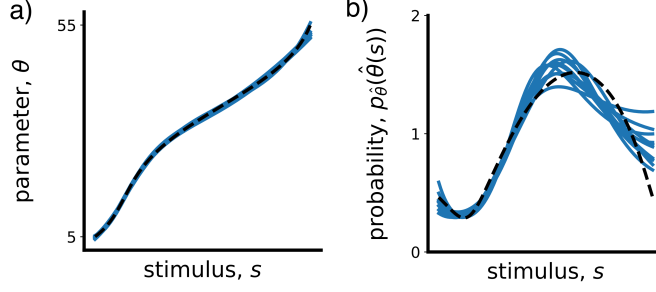

Figure 3: Fitting (a) $\theta(s)$ and (b) flat space distribution, $p_{\hat{\theta}}(\hat{\theta}(s))$ to simulated activity data (black dashes). Shown are 10 fits to the optimal solution with uniform stimulus distribution, Poisson activity, $\theta \in [5, 55]$, $f(x) = -\sqrt{x}$, $C(\theta) = \sin(2\pi/50\ \theta) + 1$ and $M = 0.75$ from 2000 trials each.

## 2.5 Estimating flat-Fisher space parameter distributions from neural activity data

Each of the above tests requires access to the distribution of flat-Fisher space neural activity parameters $\hat{\theta}(s)$, sampled across stimuli. As we show here, these activity parameter distributions can in principle be recovered from neural activity data, which consists of a series of stimulus, response pairs: $\{s_i, r_i\}_i$. For this purpose we assume that the noise in the neural encoding distribution $p(r|s)$ is known, or is well-described by an exponential family distribution, $p(r|s) = \exp(\eta(s)^T T(r) - A(\eta(s)))$, where the sufficient statistics $T(r)$ are assumed to be fixed and known, and $A(\eta(s)) = \int \exp(\eta(s)^T T(r))\, dr$ is the log-normalizer. We will require gradients and Hessians of the log normalizer, which we will take to be known for our model family, or computed by sampling the moments of the sufficient statistics at fixed $\eta$ values. Given such a model of the neural activity, our task is to fit the neural activity parameter submanifold $\eta(s)$, which we do by GLM regression, modeling $\eta(s)$ by a neural network and estimating the network's weight parameters $w$ by ascending the log-likelihood gradient, given by $\frac{\partial L}{\partial w} = \left(\sum_i T(r_i) - \frac{\partial A(\eta(s_i))}{\partial \eta}\right) \frac{\partial \eta}{\partial w}$.

Once $\eta(s)$ is known, we can use it to find the flat-Fisher space parameter distribution $p_{\hat{\theta}}(\hat{\theta})$. We can do so from Eq. (10), which shows that it is sufficient to know the stimulus Fisher information and stimulus distribution to recover the flat-Fisher space parameter distribution, $p_{\hat{\theta}}(\hat{\theta}(s))$. As a proxy for stimulus Fisher information in neural activity, we use the stimulus Fisher information of the fitted encoding model, which can be estimated by forward differentiation of the model to obtain $d\eta/ds$. By transforming inputs through the stimulus CDF prior to training, $\tilde{s} = \text{CDF}_s(s)$, the flat space distribution can be obtained directly from these Jacobians (again via Eq. (10)) by $p_{\hat{\theta}}(\hat{\theta}(\tilde{s})) \propto 1/(\sqrt{(d\eta/d\tilde{s})^T I_\eta (d\eta/d\tilde{s})})$. The parameter Fisher information in our model, $I_\eta$ is given by the Hessian of the log normalizer. In Fig. 3 we illustrate feasibility of this approach in a simple example.

## 3 Discussion

The match between efficient coding predictions and behavior of adaptive codes is no accident. By requiring first-order stability, efficient coding captures feature of codes that have achieved a steady-state of adaptation. Vice verse, by probing the adaptation behaviors of neural codes, we can characterize these codes in terms of what they are optimizing for. Here we characterized a relatively broad class of efficient coding problems, but there is still much to be achieved. For example the assumptions of one-dimensional stimuli and parameter are limiting and unrealistic. Optimizing multiple parameters would allow the shape of the parameter subspace to change, rather than simply distance, raising the question whether shape or scale optimizations dominate in different forms of adaptation. Additionally, the adaptation is not always at steady-state. In natural scenes there are many different, simultaneously active timescales that the neural code may be adapting to at any given time.

## Statement of Broader Impact

This work builds toward improved understanding of the neural code, and thus, a better understanding of operation of the nervous systems and the behavior of humans and other animals. Such understanding is important for neural prostheses and may lead to novel treatment options of human brain diseases. One potential risk is that a better understanding of neural coding, particularly value coding, lends itself to abuse for human behavioral modification. This is not an immediate concern for the work presented here.

## Acknowledgments and Disclosure of Funding

We would like to thank Johannes Bill and Haim Sompolinsky for fruitful discussions.

This work was supported by funding from the National Institutes of Health (R01MH115554, J.D.), a Scholar Award in Understanding Human Cognition by the James S. McDonnell Foundation (grant# 220020462, J.D.), and a Lefler Small Grant from the Edward R. and Anne G. Lefler Center at Harvard Medical School (L.R.).

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
