[Supplementary Material]

# Supplementary Material: Adaptation Properties Allow Identification of Optimized Neural Codes

**Luke Rast, Jan Drugowitsch**

## 1 General Approach

As discussed in the main text, we assume a neural encoding model $p(\boldsymbol{r}|s)$, which specifies how neural (population) activity $\boldsymbol{r}$ depends on a one-dimensional stimulus $s$, and is parametrized by a scalar-valued, deterministic function $\theta(s)$ such that

$$p(\boldsymbol{r}|s) = p(\boldsymbol{r}|\theta(s)), \tag{1}$$

where $\theta(s)$ is increasing in $s$. We assume that the code is efficient in that it maximizes the average across $s$ of a function $f(\cdot)$ of the square root of the Fisher information in $\boldsymbol{r}$ about $s$, $I_s(s) = \mathbb{E}_{p(\boldsymbol{r}|s)}\left[\left(\frac{\mathrm{d}}{\mathrm{d}s}\log p(\boldsymbol{r}|s)\right)^2\right]$, while satisfying, on average, some constraint $h(\boldsymbol{r}) \leq M$, that is

$$\max \; \mathbb{E}_{p_s(s)}\left[f\left(\sqrt{I_s(s)}\right)\right], \qquad \text{s.t.} \quad \mathbb{E}_{p(\boldsymbol{r}|s)p_s(s)}\left[h(\boldsymbol{r})\right] \leq M. \tag{2}$$

Specific assumptions about $f(\cdot)$ and $h(\cdot)$ are discussed in the main text, and will be revisited further below.

### 1.1 Deriving the efficient coding objective in terms of $\theta(s)$

Here we derive the $\theta(s)$ optimization from our assumptions. Because $\theta$ is a deterministic remapping of $s$, an application of the chain rule shows that the Fisher information in $\boldsymbol{r}$ about $s$ is related to the Fisher information in $\boldsymbol{r}$ about $\theta$ by

$$I_s(s) = I_\theta\big(\theta(s)\big)\left(\frac{\mathrm{d}\theta}{\mathrm{d}s}\right)^2. \tag{3}$$

Note that $I_\theta(\theta) = \mathbb{E}_{p(\boldsymbol{r}|\theta)}\left[\left(\frac{\mathrm{d}}{\mathrm{d}\theta}\log p(\boldsymbol{r}|\theta)\right)^2\right]$ is determined only by features of the encoding model. Turning to the constraint,

$$\mathbb{E}_{p_s(s)p(\boldsymbol{r}|s)}[h(\boldsymbol{r})] = \mathbb{E}_{p_s(s)}\left[\int h(\boldsymbol{r})p(\boldsymbol{r}|\theta(s))\mathrm{d}\boldsymbol{r}\right] := \mathbb{E}_{p_s(s)}\left[C(\theta(s))\right], \tag{4}$$

where we have defined $C(\theta) := \mathbb{E}_{p(\boldsymbol{r}|\theta)}\left[h(\boldsymbol{r})\right]$. Finally, we made two additional coding assumptions on $\theta(s)$, namely that $\theta(s)$ is increasing

$$\frac{\mathrm{d}\theta}{\mathrm{d}s} > 0 \tag{5}$$

and that the $\theta(s)$ curve has finite extent $L_\theta$ in $\theta$ space

$$\int \frac{\mathrm{d}\theta}{\mathrm{d}s}\mathrm{d}s = \int \mathrm{d}\theta = L_\theta. \tag{6}$$

Taking the assumed objective function, Eq. (2), and (i) plugging Eq. (3) into the objective, (ii) plugging Eq. (4) into the constraint, (iii) introducing the encoding specific constraints, Eqs. (5) and (6), yields the primary optimization problem from the main text:

$$\max_{\theta(s)} \; \mathbb{E}_{p_s(s)}\left[f\left(\sqrt{I_\theta(\theta)}\frac{\mathrm{d}\theta}{\mathrm{d}s}\right)\right] \quad \text{s.t.} \quad \mathbb{E}_{p_s(s)}\left[C(\theta(s))\right] \leq M, \quad \frac{\mathrm{d}\theta}{\mathrm{d}s} > 0, \quad \int \frac{\mathrm{d}\theta}{\mathrm{d}s}\mathrm{d}s = L_\theta. \tag{7}$$

## 1.2 Assumptions about functional forms

The features of our efficient coding problem, $p_s(s)$, $f(\sqrt{I})$, $I_\theta(\theta)$, and $C(\theta)$, are allowed to be quite flexible, but are still subject to some constraints, as listed here:

- $p_s(s)$ is assumed to be continuous, and to have no zero-probability 'gaps'. This latter assumption is for convenience, and can be removed without changing the results.

- $f(\sqrt{I})$ is assumed to be continuous and monotonically increasing. Given these assumptions, we shown in the next section that $f(\cdot)$ must additionally be concave and asymptotically sublinear for the optimization-problem to be well-behaved.

- $I_\theta(\theta)$ is assumed to be continuous and non-zero (note that Fisher information is always $\geq 0$).

- $C(\theta)$ is assumed to be continuous and non-negative. Note that non-negativity follows from non-negativity of $h(\boldsymbol{r})$.

## 1.3 Concavity and sub-linearity of the objective function

The requirement that the objective function is concave and asymptotically sub-linear can be demonstrated most directly in an optimal control approach to solving the variational optimization. Here, we interpret the efficient coding problem as optimizing a control $u(s)$ on how fast $\theta$ is increasing at each stimulus, that is, $u(s) := \frac{d\theta}{ds}$. Note that in this formulation, the stimulus $s$ plays the role of time in a traditional optimal control setting. The last two constraints in Equation 7 act as a restriction on the control set, and a boundary condition for state $\theta$, respectively. For such optimal control problems, the Pontryagin maximality principle (see [1, 3]) states that, if we form the Hamiltonian

$$H\left(s, \theta(s), q(s), u(s)\right) = q(s)u(s) + f\left(u(s)\sqrt{I_\theta(\theta(s))}\right) p_s(s) + \lambda C(\theta(s))p_s(s), \qquad (8)$$

by adding to the objective function a term composed of the product between the dynamics ($d\theta/ds = u$) and a 'momentum' co-state, $q(s)$, then the optimal trajectory will be given by a local optimization of $u(s)$,

$$u^*(s) = \mathrm{argmax}_{u(s)} H(s, \theta(s), q(s), u(s)), \qquad (9)$$

in conjunction with Hamilton's equations,

$$\frac{d\theta}{ds} = \frac{\partial H}{\partial q}, \quad \frac{dq}{ds} = -\frac{\partial H}{\partial \theta}. \qquad (10)$$

Here we focus on the control maximization:

$$\max_{u(s)} \quad q(s)u(s) + f\left(u(s)\sqrt{I_\theta(\theta)}\right) p_s(s). \qquad (11)$$

Recall that $u(s) = d\theta/ds$ is strictly positive, and $f(\cdot)$ is increasing. This means that, in order to have a maximum where $u(s) \neq \infty$, $q(s)$ must be negative everywhere, and $f(\cdot)$ must increase sublinearly as $u(s) \to \infty$. Furthermore, because $u(s) \in (0, \infty)$ cannot be at the boundaries, we have a second order condition for maximality:

$$\frac{\partial^2}{\partial u^2}\left(q(s)u(s) + f\left(u(s)\sqrt{I_\theta(\theta)}\right) p_s(s)\right) = f''\left(u(s)\sqrt{I_\theta(\theta)}\right) I_\theta(\theta)p_s(s) < 0. \qquad (12)$$

The non-negativity of $I_\theta(\theta)$ and $p_s(s)$ implies that maxima are only possible in places where $f(\cdot)$ is concave. This restriction does not fully rule out non-concave objectives, for which optimal solutions could transition discontinuously between concave regions of the objective function. This behavior might be interesting, but we limit the analysis to continuous $u(s)$. That is to say, we require our solution to be well-behaved, in the sense that $\theta(s)$ is $C^1$, continuous and with continuous derivatives. This is the standard requirement for optimization by Euler-Lagrange. Hence, in order for solutions $\theta(s)$ to be $C^1$, the objective function must be concave and asymptotically sublinear as its argument increases.

## 2  Finding efficient coding solutions

Here we derive the solution to the efficient coding problem above. We first outline the general approach, leading to a system of ordinary differential equations that describe the efficient coding solution. From this we derive in turn the three special-case analytic solution described in the main text.

### 2.1  General optimal solution

There are several ways to solve the general optimization problem. Here we walk through an approach that is particularly intuitive, but requires some setup. Start with the general optimization problem (Eq. (7)),

$$\max_{\theta(s)} \mathbb{E}_{p_s(s)}\left[ f\left( \sqrt{I_\theta(\theta)}\frac{d\theta}{ds}\right)\right] \quad \text{s.t.} \quad \mathbb{E}_{p_s(s)}[C(\theta)] \le M, \quad \frac{d\theta}{ds} > 0, \quad \int \frac{d\theta}{ds}ds = L_\theta. \quad (13)$$

Because $\theta(s)$ is continuous and strictly increasing, we know that $\theta$ is an invertible function of $s$, and can therefore be re-expressed in terms of cumulative distribution functions or probability density functions

$$\theta(s) = \mathrm{CDF}_\theta^{-1}(\mathrm{CDF}_s(s)); \qquad p_\theta(\theta)\mathrm{d}\theta = p_s(s)\mathrm{d}s. \quad (14)$$

Equivalently,

$$s(\theta) = \mathrm{CDF}_s^{-1}(\mathrm{CDF}_\theta(\theta)); \qquad \frac{\mathrm{d}\theta}{\mathrm{d}s} = \frac{p_s(s(\theta))}{p_\theta(\theta)}. \quad (15)$$

The idea here is to transform the optimization of the function $\theta(s)$ into an optimization of $p_\theta(\theta)$, the probability density function. To do this, we first change the variable of integration from $p_s(s)\mathrm{d}s$ to $p_\theta(\theta)\mathrm{d}\theta$ and substitute the ratio of probability density functions in place of the $\theta$ derivative,

$$\max_{p_\theta(\theta)} \int_0^{L_\theta} f\left( \frac{p_s(s(\theta))}{p_\theta(\theta)}\sqrt{I_\theta(\theta)}\right) p_\theta(\theta)\mathrm{d}\theta \quad \text{s.t.} \quad \int_0^{L_\theta} C(\theta)p_\theta(\theta)\mathrm{d}\theta \le M, \quad \int_0^{L_\theta} p_\theta(\theta)\mathrm{d}\theta = 1 \quad (16)$$

In this formulation, the final two (parameter) constraints have been absorbed into assumptions on the probability density function. First, $p_\theta(\theta)$ is positive. Second, the length of $\theta$-space is $L_\theta$, and $p_\theta$ is normalized over this length. Here we have WLOG set the lower bound of $\theta$ to zero. Note that $s(\theta) = \mathrm{CDF}_s^{-1}(\mathrm{CDF}_\theta(\theta))$ has its own $\theta$ dependence, which introduces some notational complexity. To handle this complexity, we define the additional functions $\alpha(\theta)$ and $\pi_s(\alpha)$, given by

1. $\mathrm{CDF}_\theta(\theta) = \alpha(\theta)$,

2. $\frac{d\alpha}{d\theta} = p_\theta(\theta)$ ,

3. $p_s(s(\theta)) = p_s(\mathrm{CDF}_s^{-1}(\mathrm{CDF}_\theta(\theta))) = \pi_s(\alpha(\theta))$ .

This turns the optimization problem into

$$\max_{\alpha(\theta),p_\theta(\theta)} \int_0^{L_\theta} f\left( \frac{\sqrt{I_\theta(\theta)}\pi_s(\alpha(\theta))}{p_\theta(\theta)}\right) p_\theta(\theta)\mathrm{d}\theta \quad \text{s.t.} \quad \begin{cases} \int C(\theta)p_\theta(\theta)\mathrm{d}\theta \le M, \\ \int p_\theta(\theta)\mathrm{d}\theta = 1, \\ \frac{d\alpha}{d\theta} = p_\theta(\theta) \end{cases} \quad (17)$$

Here we are optimizing both of the functions $\alpha(\theta)$ and $p_\theta(\theta)$ under the constraint that $\frac{d\alpha}{d\theta} = p_\theta(\theta)$ (by the definition of $\alpha(\theta)$), that is, that they specify the cumulative and probability density functions of the same distribution. The $p_\theta(\theta)$ optimization has a familiar form. Along with the top two constraints, it takes the from of a constrained minimization of the $-f$ divergence [2] between $p_\theta$ and $\sqrt{I_\theta(\theta)}\pi_s(\alpha(\theta))$. In this problem, however, the target distribution is also dependent on the distribution that we are optimizing. We will see that this target adaptation is what accounts for stimulus-distribution dependent adaptation.

Optimizing both $\alpha(\theta)$ and $p_\theta(\theta)$, the Euler-Lagrange equations result in the system of ordinary differential equations,

$$\hat{f}\left(\frac{\sqrt{I_\theta(\theta)}\pi_s(\alpha(\theta))}{p_\theta(\theta)}\right) = \lambda C(\theta) + Z - \gamma(\theta) \tag{18}$$

$$\frac{d\gamma}{d\theta} = f'\left(\frac{\sqrt{I_\theta(\theta)}\pi_s(\alpha(\theta))}{p_\theta(\theta)}\right)\sqrt{I_\theta(\theta)}\pi_s'(\alpha(\theta)), \tag{19}$$

$$\frac{d\alpha}{d\theta} = p_\theta(\theta). \tag{20}$$

Here, $\lambda$, $Z$, and $\gamma(\theta)$ are Lagrange multipliers in charge of enforcing, respectively, the constraint on $C(\theta)$, normalization of $p_\theta(\theta)$, and the equality between $p_\theta$ and the derivative of the $\theta$ CDF, $\alpha(\theta)$. The constraint on $C(\theta)$, and the required normalization, can be satisfied by solving (numerically if necessary) for the values of $\lambda$ and $Z$. The objective function appears here in two forms: the derivative, $f'(\cdot)$, in Eq. (19), and the Legendre transform, $\hat{f}(\cdot)$, in Eq. (18), which is defined as $\hat{f}(x) = f'(x)x - f(x)$.

## 2.2 Special-case analytic solutions

We now derive each of the special cases in turn.

### 2.2.1 Log-objective function, $f(\sqrt{I}) \propto \log I$

Setting $f(x) = \log x$, implies that $f'(x) = 1/x$, and $\hat{f}(x) = 1 - \log x$, in which case the $\gamma(\theta)$ dynamics, Eq. (19), become

$$\frac{d\gamma}{d\theta} = \frac{\pi_s'(\alpha(\theta))}{\pi_s(\alpha(\theta))}p_\theta(\theta) = \frac{d}{d\theta}\log\pi_s(\alpha(\theta)) \tag{21}$$

Hence, $\gamma(\theta) = \log\pi_s(\alpha(\theta)) + c_0$ with constant $c_0$. Plugging this result into Eq. (18) cancels the $\pi_s$ dependence on the right-hand side and, absorbing constants into $Z$, the remaining equation can be solved to yield

$$p_\theta(\theta) = Z^{-1}\sqrt{I_\theta(\theta)}\exp(-\lambda C(\theta)). \tag{22}$$

### 2.2.2 Unconstrained optimization, $M \to \infty$

For convenience denote $G(\theta) = \frac{\sqrt{I_\theta(\theta)}\pi_s(\alpha(\theta))}{p_\theta(\theta)}$. In Eq. (18), setting $C(\theta) = 0$ (or $\lambda = 0$ if eg. a constraint is present, but not binding) means that $\gamma(\theta) = Z - \hat{f}(G(\theta))$. Furthermore, note that from the definition $\hat{f}(x) = f'(x)x - f(x)$,

$$\frac{d\gamma}{d\theta} = -\frac{d}{d\theta}\hat{f}(G(\theta)) = -f''(G(\theta))G(\theta)G'(\theta). \tag{23}$$

This $\gamma$ derivative can be plugged directly into Eq. (19), yielding:

$$-\frac{f''(G(\theta))}{f'(G(\theta))}G'(\theta) = \frac{\pi_s'(\alpha)}{\pi_s(\alpha)}p_\theta(\theta) \tag{24}$$

Equivalently,

$$\frac{d}{d\theta}\log f'(G(\theta)) = \frac{d}{d\theta}\log\frac{1}{\pi_s(\alpha(\theta))}. \tag{25}$$

Hence,

$$f'\left(\frac{\sqrt{I_\theta(\theta)}\pi_s(\alpha(\theta))}{p_\theta(\theta)}\right) \propto \frac{1}{\pi_s(\alpha(\theta))}, \tag{26}$$

from which it follows that

$$p_\theta(\theta) = \sqrt{I_\theta(\theta)}\frac{p_s(s(\theta))}{f'^{-1}\left(\frac{Z}{p_s(s(\theta))}\right)}, \tag{27}$$

the form shown in the main text.

### 2.2.3 Uniform stimulus probability density, $p_s(s) \propto 1$

From the discussion above, when the stimulus distribution is uniform, that is $p_s(s) = p_s$, and $\pi_s(\cdot) \propto 1$, the $-f$ divergence interpretation becomes clear. In this case, $\pi'_s(\alpha(\theta)) = 0$, making $\gamma(\theta)$ a constant that can be absorbed into $Z$. Then, solving Eq. (18) yields

$$p_\theta(\theta) = \sqrt{I_\theta(\theta)} \frac{p_s}{\hat{f}^{-1}(\lambda C(\theta) + Z)}. \tag{28}$$

### 2.3 Multiple constraints

In the above derivation, the constraint $C(\theta)$ passes through unchanged to give a factor of $\lambda C(\theta)$ in Eq. (18). The same will occur with multiple constraints, to give a term $\boldsymbol{\lambda}^T \boldsymbol{C}(\theta)$ in place of $\lambda C(\theta)$ in all of the above solutions. Satisfying the constraints then requires solving a higher dimensional system of equations for the values of the Lagrange multipliers, but the results remain qualitatively unchanged.

## 3 Flat-Fisher space reparameterization

As discussed in the main text, transformation to the flat-Fisher space facilitates an understanding of the efficient coding solutions that is invariant to our parameterization of the neural activity. Here we derive this remapping, apply it to special-case solutions and the overall problem formulation, and then derive the relationship between the flat-Fisher space parameter distribution and the Fisher information in stimulus space.

### 3.1 Defining the reparametrization

We want our reparameterization to satisfy

$$\mathrm{d}\hat{\theta} = \sqrt{I_\theta(\theta)}\mathrm{d}\theta. \tag{29}$$

First we have to ensure that such a reparameterization is well behaved and well defined. Because $\sqrt{I_\theta(\theta)}$ is strictly positive ("strictly" by assumption), we can always solve this differential equation to find a strictly monotonic (increasing) function $F$:

$$\hat{\theta}(\theta) = F(\theta) = \int_{\theta_0}^{\theta} \sqrt{I_\theta(\theta')}\mathrm{d}\theta'. \tag{30}$$

This reveals that such a reparameterization will be invertible and thus well behaved. However, the transformation is only defined up to the integration constant. In fact, the Fisher information can be made constant by any transformation of the form $\mathrm{d}\hat{\theta} = a\sqrt{I_\theta(\theta)}\mathrm{d}\theta$ with constant $a$:

$$I_{\hat{\theta}}\left(\hat{\theta}\right) = I_\theta\left(\theta\left(\hat{\theta}\right)\right)\left(\frac{\mathrm{d}\theta}{\mathrm{d}\hat{\theta}}\right)^2 = \frac{1}{a^2} \tag{31}$$

Therefore, the flat-Fisher space is well-defined only up to shifting the origin and re-scaling the length. For our purposes, neither of these will matter: the scale only serves to change the normalization of the distribution $p_{\hat{\theta}}$, and we are free to choose the origin of $\hat{\theta}$.

### 3.2 Reparameterization of special-case solutions

As noted in the main text, all of the special-case solutions are scaled by a factor of $\sqrt{I_\theta}$. Owing to the invertible mapping between $\theta$ and $\hat{\theta}$, this scaling can be removed by reparameterization to the flat-Fisher space,

$$p_{\hat{\theta}}\left(\hat{\theta}\right) = p_\theta\left(\theta\left(\hat{\theta}\right)\right)\frac{\mathrm{d}\theta}{\mathrm{d}\hat{\theta}} = \frac{p_\theta\left(\theta\left(\hat{\theta}\right)\right)}{\sqrt{I_\theta\left(\theta\left(\hat{\theta}\right)\right)}}. \tag{32}$$

This removes $I_\theta$ terms from the solution, but does require reparameterizing the constraint to $C\left(\theta\left(\hat{\theta}\right)\right) = C\left(F^{-1}\left(\hat{\theta}\right)\right) := C_{\hat{\theta}}\left(\hat{\theta}\right)$.

### 3.3 Reparameterization of problem formulation

Explicit $\sqrt{I_\theta}$ dependence can be fully removed from the optimization problem by reparameterizing prior to solving. We first note that

$$\frac{d\hat{\theta}}{ds} = \frac{d\hat{\theta}}{d\theta}\frac{d\theta}{ds} = \sqrt{I_\theta}\frac{d\theta}{ds}. \tag{33}$$

and that

$$\frac{d\hat{\theta}}{ds} = \frac{p_s(s)}{p_{\hat{\theta}}(\hat{\theta})} = \frac{\sqrt{I_\theta(\theta)}p_s(s)}{p_\theta(\theta)}. \tag{34}$$

From Eq. (33) and the fact that $\sqrt{I_\theta}$ is strictly positive, it follows that, $d\theta/ds > 0 \iff d\hat{\theta}/ds > 0$. Thus, the positivity constraint on the $\theta$ derivatives can be replaced by a positivity constraint on $\hat{\theta}$ derivatives, and, plugging Eq. (33) in the objective function in the optimization, Eq. (7), yields

$$\max_{\hat{\theta}(s)} \mathbb{E}_{p_s(s)}\left[f\left(\frac{d\hat{\theta}}{ds}\right)\right] \quad \text{s.t.} \quad \mathbb{E}_{p_s(s)}\left[C_{\hat{\theta}}\left(\hat{\theta}\right)\right] \leq M, \quad \frac{d\hat{\theta}}{ds} > 0, \quad \int \frac{d\theta}{ds}ds = L_\theta. \tag{35}$$

Equation 34 ensures that the solution approach described above goes through unchanged, replacing $\sqrt{I_\theta(\theta)}/p_\theta(\theta)$ by $1/p_{\hat{\theta}}(\hat{\theta})$.

One step remains for transformation to the flat-Fisher space: satisfaction of the length constraint on $\theta$. The idea here is to make use of the invertible mapping $\theta \leftrightarrow \hat{\theta}$, such that if we move from a starting location in $\theta$-space, $\theta_0$, by some distance $L_\theta$, then, in doing so, we will have moved from the corresponding location in $\hat{\theta}$-space, $\hat{\theta}_0$, by a distance $L_{\hat{\theta}}$. Regardless of the value of $L_\theta$, the invertible mapping provides the transformation from $\theta$-distances to $\hat{\theta}$-distances. More specifically,

$$L_\theta = \int_{s_0}^{s_1} \frac{d\theta}{ds}ds = \int_{\theta_0}^{\theta_1} d\theta = \theta_1 - \theta_0, \tag{36}$$

while

$$L_{\hat{\theta}} = \int_{s_0}^{s_1} \frac{d\hat{\theta}}{ds}ds = \int_{\theta_0}^{\theta_1} \sqrt{I_\theta(\theta)}d\theta = F(\theta_1) - F(\theta_0). \tag{37}$$

So, given a starting point, $\theta_0$ and a $\theta$-length, $L_\theta$, we can find the corresponding $\hat{\theta}$-length by:

$$L_{\hat{\theta}} = F(\theta_0 + L_\theta) - F(\theta_0). \tag{38}$$

The same logic provides a transformation from $\hat{\theta}$-lengths to $\theta$-lengths:

$$L_\theta = F^{-1}(\hat{\theta}_0 + L_\theta) - F^{-1}(\hat{\theta}_0). \tag{39}$$

Thus, we can transform uniquely between lengths $L_\theta$ and $L_{\hat{\theta}}$ so that the $\theta$ length constraint will be satisfied if and only if the $\hat{\theta}$ length constraint is satisfied.

In fact, because the $\hat{\theta}$ length gives a constraint on the integrated root-Fisher information, it can be transformed into stimulus space using $\sqrt{I_s}ds = \sqrt{I_\theta}d\theta$, to give:

$$L_{\hat{\theta}} = \int_{\theta_0}^{\theta_1} \sqrt{I_\theta(\theta)}d\theta = \int_S \sqrt{I_s(s)}ds. \tag{40}$$

This is the total root-Fisher constraint of earlier works [4, 5, 6] and, because it depends on the stimulus Fisher information, it is independent of how we choose to parameterize the problem.

### 3.4 Flat-Fisher distribution depends only on observable quantities

Here we derive the relationship between the flat-Fisher parameter distribution $p_{\hat{\theta}}(\hat{\theta})$ and two observable (or controllable) quantities: the stimulus distribution $p_s(s)$ and the Fisher information in the neural activity about the stimulus $I_s(s)$. The flat-Fisher parameter $\hat{\theta}$ is a monotonic increasing function of the parameter $\theta$, and thus a monotonic increasing function of the stimulus $s$. So,

$$p_{\hat{\theta}}\left(\hat{\theta}(s)\right) = p_s(s) \,/\, \frac{d\hat{\theta}}{ds}. \tag{41}$$

Because $\mathrm{d}\hat{\theta} = a\sqrt{I_\theta(\theta)}\mathrm{d}\theta$ and $I_s(s) = I_\theta(\theta(s))\frac{\mathrm{d}\theta}{\mathrm{d}s}^2$, we have

$$\frac{d\hat{\theta}}{ds} = \frac{d\hat{\theta}}{d\theta}\frac{d\theta}{ds} = a\sqrt{I_\theta(\theta(s))}\frac{\sqrt{I_s(s)}}{\sqrt{I_\theta(\theta(s))}}. \tag{42}$$

Hence,

$$\boxed{p_{\hat{\theta}}\left(\hat{\theta}(s)\right) = \frac{1}{a}\frac{p_s(s)}{\sqrt{I_s(s)}}.} \tag{43}$$

This can also be derived heuristically by taking the ratio of two changes of variable:

$$p_s(s)ds = p_\theta(\theta)d\theta = p_{\hat{\theta}}(\hat{\theta})d\hat{\theta}, \tag{44}$$

$$\sqrt{I_s(s)}ds = \sqrt{I_\theta(\theta)}d\theta = \sqrt{I_{\hat{\theta}}(\hat{\theta})}d\hat{\theta} = \frac{1}{a}d\hat{\theta}, \tag{45}$$

to get:

$$\frac{p_s(s)}{\sqrt{I_s(s)}} = \frac{p_\theta(\theta)}{\sqrt{I_\theta(\theta)}} = ap_{\hat{\theta}}(\hat{\theta}). \tag{46}$$

Thus,

$$p_{\hat{\theta}}(\hat{\theta}(s)) \propto \frac{p_\theta(\theta)}{\sqrt{I_\theta(\theta)}} = \frac{p_s(s)}{\sqrt{I_s(s)}}. \tag{47}$$

# 4   Adaptation of optimal codes to stimulus distribution

## 4.1   Stimulus distribution invariance implies log objective function

It is difficult to analyze the role of the stimulus distribution in the general efficient coding solutions that we have presented so far (Eqs. (18)-(20)) because the stimulus distribution comes into play in two places: in the argument to the objective function and in the $\gamma$ dynamics. Here we instead use a different approach to solving the optimization problem. Specifically, we reparameterize the parameter $\theta$ to the flat-Fisher space $\hat{\theta}$ *and* we reparameterize the stimulus to the space $\hat{s} = \mathrm{CDF}_s(s)$ in which the $\hat{s}$ probability is uniform. This means that stimulus density terms become $p_s(\mathrm{CDF}_s^{-1}(\hat{s})) = \pi_s(\hat{s})$ and that

$$\frac{\mathrm{d}\hat{\theta}}{\mathrm{d}\hat{s}} = \frac{1}{p_{\hat{\theta}}(\hat{\theta}(\hat{s}))}. \tag{48}$$

Now we solve the optimization by optimal control (see also Sec. (1.3)) keeping $\hat{s}$ as the independent variable. Denoting $u = \frac{\mathrm{d}\hat{\theta}}{\mathrm{d}\hat{s}}$ gives the Hamiltonian

$$H\left(\hat{s}, \hat{\theta}(\hat{s}), q(\hat{s}), u(\hat{s})\right) = q(\hat{s})u(\hat{s}) + f\left(u(\hat{s})\pi_s(\hat{s})\right) + \lambda C_{\hat{\theta}}\left(\hat{\theta}(\hat{s})\right). \tag{49}$$

Applying the Pontryagin maximality principle (see Sec.(1.3) and [1, 3]) and rearranging the outputs gives

$$\boxed{\lambda C'_{\hat{\theta}}\left(\hat{\theta}(\hat{s})\right) = \frac{\mathrm{d}}{\mathrm{d}\hat{s}}f'\left(u(\hat{s})\pi_s(\hat{s})\right)\pi_s(\hat{s}),} \tag{50}$$

a second order differential equation for $\hat{\theta}(\hat{s})$.

Let us first examine the constraint free case. Setting $C'_{\hat{\theta}}(\hat{\theta}) = 0$, the differential Equation (50) can be integrated to give $f'(u\pi_s)\pi_s = \mathrm{const}$. Thus, the $\pi_s$ dependence will only cancel if $f'(x) \propto 1/x$, in which case a uniform solution (unconstrained, log objective) is achieved. Otherwise $u(\hat{s})$ will be a non-trivial function of $\pi_s(\hat{s})$ and stimulus distribution dependence will be assured. Thus, when constraints are absent, a log objective function is required for stimulus distribution independence.

In the presence of constraints, we can make a similar argument on the full differential Equation (50). Here the dependence on $\pi_s(\hat{s})$ will again cancel if and only if $f$ is a logarithm so that $f' \propto 1/x$. Otherwise, the differential equation that determines $u(\hat{s}) = \mathrm{d}\hat{\theta}/\mathrm{d}\hat{s} = 1/p_{\hat{\theta}}(\hat{\theta}(\hat{s}))$ will depend on the probability density of the stimulus. This means that each stimulus distribution, $\pi_{si}$, will have

its own differential equation that determines the solution. Some of these differential equations will share solutions, for example stimulus distributions that are shifted from each other by a constant stimulus offset. We need to show that, unless the $\pi_s$ dependence explicitly cancels, not *all* of the different $\pi_{si}$ differential equations will share the *same* solution. This may be fairly obvious, but we give an argument here that also foreshadows the results of the next section.

Suppressing the $\hat{s}$ function arguments, assume that for a given stimulus distribution, $\pi_{s1}$, $u_1$ is the solution to the differential equation:

$$\lambda_1 C'_{\hat{\theta}}\left(\hat{\theta}_1\right) = \frac{\mathrm{d}}{\mathrm{d}\hat{s}} f'\left(u_1 \pi_{s1}\right)\pi_{s1}. \tag{51}$$

Because $\hat{\theta}(\hat{s})$ and its inverse map one bounded space to another, the function $1/u_1(\hat{s})$ can be normalized to a probability density and we can probe the optimal solution using this probability density $\pi_{s2} = A/u_1$ ($A$ constant). If the parameter distribution is adaptation invariant, the two stimulus distributions $\pi_{s1}$ and $\pi_{s2}$ will have the same solution: $u_1 = u_2$. This requires

$$\frac{\mathrm{d}}{\mathrm{d}\hat{s}} f'\left(u_1 \pi_{s1}\right)\pi_{s1} = \frac{\mathrm{d}}{\mathrm{d}\hat{s}} f'\left(u_2 \frac{A}{u_1}\right)\frac{A}{u_1} = f'\left(A\right)\frac{\mathrm{d}}{\mathrm{d}\hat{s}}\frac{A}{u_1}. \tag{52}$$

Thus, invariance requires

$$\frac{\mathrm{d}}{\mathrm{d}\hat{s}} f'\left(u_1 \pi_{s1}\right)\pi_{s1} \propto \frac{\mathrm{d}}{\mathrm{d}\hat{s}}\frac{1}{u_1}, \tag{53}$$

for all probability distributions $\pi_{s1}$. Integrating both sides, we see that this is only possible if (i) $f'(x) \propto 1/x$ and so $f$ is log, or (ii) $1/u_1 \propto \pi_{s1}$. In this second case, we have a 'fixed point' of the mapping because the distribution of parameters is proportional to the stimulus distribution, this is examined more below. For our current purpose, this proportionality must hold for all stimulus distributions $\pi_s$. However, this evidently requires that different stimulus distribution result in different functions $u$ and so cannot lead to stimulus distribution invariance. Thus, regardless of constraints, stimulus distribution invariance will only hold if the objective function is logarithmic.

## 4.2 Derivation of the adaptation fixed point

We have derived both the general solution to the efficient coding optimization and the transformation to the flat-Fisher space. We can now find the fixed point of neural adaptation in the flat-Fisher space. First, we rewrite the optimization solutions, Eqs. (18)-(20) in the flat-Fisher space, making use of Eq. (34),

$$\hat{f}\left(\frac{\pi_s\left(\hat{\alpha}\left(\hat{\theta}\right)\right)}{p_{\hat{\theta}}\left(\hat{\theta}\right)}\right) = \lambda C_{\hat{\theta}}\left(\hat{\theta}\right) + Z - \gamma\left(\hat{\theta}\right), \tag{54}$$

$$\frac{\mathrm{d}\gamma}{\mathrm{d}\hat{\theta}} = f'\left(\frac{\pi_s\left(\hat{\alpha}\left(\hat{\theta}\right)\right)}{p_{\hat{\theta}}\left(\hat{\theta}\right)}\right)\pi'_s\left(\hat{\alpha}\left(\hat{\theta}\right)\right), \tag{55}$$

$$\frac{\mathrm{d}\hat{\alpha}}{\mathrm{d}\hat{\theta}} = p_{\hat{\theta}}\left(\hat{\theta}\right). \tag{56}$$

Here $\hat{\alpha}$ is now the cumulative distribution function of $p_{\hat{\theta}}$, while the stimulus distribution dependent term $\pi_s(\cdot)$ remains unchanged.

We search for a fixed point in the sense of distributions: after adapting to a particular distribution of stimuli, $p_s$, the neural code will be characterized by a distribution of flat-space parameters, $p_{\hat{\theta}}$. When the 'input' stimulus distribution is the same as the 'output' parameter distribution, this can be said to be an adaptation fixed point. That is,

$$p_{\hat{\theta}}\left(\hat{\theta}\right) \propto p_s\left(s\left(\hat{\theta}\right)\right) = \pi_s\left(\alpha\left(\hat{\theta}\right)\right). \tag{57}$$

Proportionality allows the stimulus domain and parameter domain to have different lengths and, as we saw above, $p_{\hat{\theta}}(\hat{\theta})$ is only defined up to a proportionality constant due to rescaling of the $\hat{\theta}$ domain.

It may seem strange to compare distributions of stimuli to distributions of activity parameters, but it can be seen by integrating $p_s(s)ds = p_{\hat{\theta}}(\hat{\theta})d\hat{\theta}$ that, at such a fixed point,

$$\hat{\theta}(s) = as + b, \tag{58}$$

with $a$ and $b$ constants. Hence, $\hat{\theta}$ and $s$ are related through a linear scaling and shifting. Because $\hat{\theta}$ is only defined up to such a scaling and shifting, this is as close as we can come to identifying stimuli and (flat space) parameters. Intuitively, the distribution $p_{\hat{\theta}}$ is shaped by two factors: (i) the likelihood of the stimulus that each value of $\hat{\theta}$ encodes, and (ii) the degree to which the codes for nearby stimuli are stretched apart or compressed together in parameter space. At the fixed point, there will be no stretching or compression, so that the parameter probability density is determined exclusively by the probability density of the stimulus encoded.

At such a fixed point, Equation (57) shows that the argument to $\hat{f}$ in Eq. (54), and the argument to $f'$ in Eq. (55) become constant,

$$\frac{\pi_s\left(\alpha\left(\hat{\theta}\right)\right)}{p_{\hat{\theta}}\left(\hat{\theta}\right)} = c_0. \tag{59}$$

Taking Eq. (55) and multiplying the right-hand side of by $c_0$ and also dividing it by the equivalent ratio of probability densities from Eq. (59) gives:

$$\frac{d\gamma}{d\theta} = f'(c_0)\pi_s'\left(\alpha\left(\hat{\theta}\right)\right)c_0\frac{p_{\hat{\theta}}\left(\hat{\theta}\right)}{\pi_s\left(\alpha\left(\hat{\theta}\right)\right)} = c_1\frac{d}{d\theta}\log\pi_s\left(\alpha\left(\hat{\theta}\right)\right) \tag{60}$$

This is a constant multiple of the log-objective function solution in Eq. (21), and can be substituted into Eq. (54) to give (combining additive constants into $Z$ and multiplicative constants in $\lambda$ and $Z$)

$$\log\pi_s\left(\alpha\left(\hat{\theta}\right)\right) = \lambda C_{\hat{\theta}}\left(\hat{\theta}\right) + Z. \tag{61}$$

Hence,

$$p_{\hat{\theta}}\left(\hat{\theta}\right) \propto \pi_s\left(\alpha\left(\hat{\theta}\right)\right) \propto \exp\left(\lambda C_{\hat{\theta}}\left(\hat{\theta}\right)\right). \tag{62}$$

Because it is a function of the parameter directly, the constraint function, $C_{\hat{\theta}}(\hat{\theta})$ will be dependent on the particular parameterization that is used, here the flat-Fisher space parameterization. If required, it can be converted back to $\theta$-space given knowledge of $I_\theta(\theta)$ via the integrated transformation from above, $C(\theta) = C_{\hat{\theta}}(F(\theta))$.

Once we know $C_{\hat{\theta}}(\hat{\theta})$, we can also recover the objective function. Probing with a uniform stimulus distribution, we know that the resulting flat-space distribution follows

$$p_{\hat{\theta}}(\hat{\theta}) = \frac{p_s}{\hat{f}^{-1}(\lambda C_{\hat{\theta}}(\hat{\theta}) + Z)}. \tag{63}$$

This allow us to fit $\hat{f}^{-1}$ by regressing $C_{\hat{\theta}}(\hat{\theta}(s))$ against $1/p_{\hat{\theta}}(\hat{\theta}(s))$. This does require determining the function $\hat{\theta}(s)$ for the code adapted to the uniform stimulus distribution, which can be accomplished by integrating (numerically) $1/p_{\hat{\theta}}(\hat{\theta}(s))ds$.

## 5    Estimating the distribution of neural activity parameters from data

Because the distribution $p_{\hat{\theta}}(\hat{\theta}(s))$ is crucial for characterization of the objective and constraint functions that shape a neural code, we showed in a main text a proof of principle for recovery of $p_{\hat{\theta}}(\hat{\theta}(s))$ from neural activity data. Here we present the details of the model architecture and training used.

### 5.1    Parameter regression

The first step in recovery of the flat space parameter distribution from activity data, $\{s_i, r_i\}_i$, is regression from stimuli to the parameters underlying the neural activity. Given a known or identified

neural activity model, $p(\boldsymbol{r}|\boldsymbol{\eta}(s)) = \exp(\boldsymbol{\eta}(s)^T \boldsymbol{T}(\boldsymbol{r}) - A(\boldsymbol{\eta}(s)))$, we aim to find the function $\boldsymbol{\eta}(s)$ by regression. Here, we use a neural network to perform maximum likelihood regression. Parameterizing $\boldsymbol{\eta}(s)$ by weights, $\boldsymbol{w}$, we can compute gradients of the log likelihood of the data under the model according to:

$$\frac{\partial L}{\partial \boldsymbol{w}} = \left( \sum_i \boldsymbol{T}(\boldsymbol{r}_i) - \frac{\partial A(\eta(s_i))}{\partial \boldsymbol{\eta}} \right) \frac{\partial \boldsymbol{\eta}}{\partial \boldsymbol{w}}, \tag{64}$$

where $L$ is the log-likelihood of the activity data, $L = \sum_i \log p\left(\boldsymbol{r}_i | \boldsymbol{\eta}(s_i)\right)$. Here $\boldsymbol{T}(\boldsymbol{r})$ are the sufficient statistics and $\partial A / \partial \boldsymbol{\eta}$ is the parameter to moment mapping of our activity model. Concretely, the term in parentheses is calculated from the data and model output, and used as the initial gradient for backpropagation to produce the weight gradients.

For the example in the main text, we used a neural network with 4 layers of width 3 and $\texttt{tanh}$ non-linearity implemented in PyTorch. We trained on 2k data points with 20k repeats of the training set and mini-batches of size 100 using the Adam optimizer with learning rate, $10^{-3}$ and regularization by weight decay with strength $10^{-2}$. Training hyperparameters and network architecture were set manually based on recovery of a validation $\boldsymbol{\eta}(s)$ curve (different from the test curve) under the same (Poisson noise) loss function. Hyperparameters were probed within a 10-fold range for training hyperparameters and a 2-fold range for architecture parameters.

## 5.2 Extracting the flat-Fisher distribution

Having performed the parameter regression, we now want to estimate the parameter distribution in the flat-Fisher space. This can be accomplished using Eq. (43),

$$p_{\hat{\theta}}\left(\hat{\theta}(s)\right) \propto \frac{p_s(s)}{\sqrt{I_s(s)}}. \tag{65}$$

We know $p_s(s)$, so all that is left is to find the stimulus Fisher information, $I_s(s)$. Here, we use the Fisher information in our trained model of neural activity as a proxy for the stimulus Fisher information in the neural activity. With the parameter mapping $\boldsymbol{\eta}(s)$ from our trained model, this is given by:

$$I_s(s) = \frac{d\boldsymbol{\eta}}{ds}^T I_{\boldsymbol{\eta}}(\boldsymbol{\eta}(s)) \frac{d\boldsymbol{\eta}}{ds}. \tag{66}$$

The $\boldsymbol{\eta}$ derivatives can be computed by forward differentiation of the fit $\boldsymbol{\eta}(s)$ function. The model Fisher information function, $I_{\boldsymbol{\eta}}(\cdot)$, is specified by our exponential family model of the noise in neural activity, we take the function to be known and evaulated on fit outputs $\boldsymbol{\eta}(s)$. Alternatively, it could be sampled, using the identity $I_{\boldsymbol{\eta}}(\boldsymbol{\eta}) = \text{cov}_{p(\boldsymbol{r}|\boldsymbol{\eta})}[\boldsymbol{T}(\boldsymbol{r})]$.

Instead of computing $I_s$ and dividing the result out from the (known) stimulus distribution, we can also CDF transform the incoming data prior to training, $\tilde{s} = \text{CDF}_s(s)$ so that the output parameters are $\boldsymbol{\eta}(\tilde{s})$. In this case, the parameter derivatives are:

$$\frac{d\boldsymbol{\eta}}{d\tilde{s}} = \frac{d\boldsymbol{\eta}}{ds} \frac{ds}{d\tilde{s}} = \frac{d\boldsymbol{\eta}}{ds} \frac{1}{p_s(s(\tilde{s}))}. \tag{67}$$

Then,

$$I_{\tilde{s}} = \frac{d\boldsymbol{\eta}}{d\tilde{s}}^T I_{\boldsymbol{\eta}}(\boldsymbol{\eta}(\tilde{s})) \frac{d\boldsymbol{\eta}}{d\tilde{s}} = \frac{d\boldsymbol{\eta}}{ds}^T I_{\boldsymbol{\eta}}(\boldsymbol{\eta}(\tilde{s})) \frac{d\boldsymbol{\eta}}{ds} \frac{1}{p_s(s(\tilde{s}))^2} = \frac{I_s(s(\tilde{s}))}{p_s(s(\tilde{s}))^2}. \tag{68}$$

Hence,

$$p_{\hat{\theta}}(\hat{\theta}(s)) \propto \frac{1}{\sqrt{I_{\tilde{s}}(\text{CDF}_s(s))}}. \tag{69}$$

The constant of proportionality can be found by normalization of this function.