[Reviews · NeurIPS 2020]

Review 1

Summary and Contributions: The authors explore optimal, efficient neural codes based on Fisher information of a one-dimensional stimulus. The goal of the paper was to build a general framework to help discover what objective functions neural codes might be optimal for (rather than requiring a hand-selected value).

Strengths: The motivations and setup were well-defined, and much of the theory was well-grounded. In particular, it's a compelling notion to discover what neural codes are doing rather than looking for specific pre-defined optimalities. The generality of the framework may be of interest to those in the NeurIPS community studying neural coding.

Weaknesses: Unfortunately, I found much of the results and methods difficult to follow. In particular, the instructions for applying this methodology to real data was not obvious (sections 2.4-2.5) and lacked a concrete example. For example, how would perform the fixed-point search on line 255? (and when does such a distribution exist?) If this requires a parametric assumption, is in the Poisson case in section 2.5, what is gained by the general framework that does not require these assumptions (line 106-109)? UPDATE: The authors' response has helped clarify some of my questions about the fixed point approach and the modeling assumptions.

Correctness: The approach and methodology used is appropriate and appears correct.

Clarity: Most of the paper is well-written, though I found some of the derivations and presentations of the results unclear. The contents of figure 2 and the caption could have been more clearly labeled and explained to better demonstrate the coding framework. (For 2b, does the “uniform” part only apply to the left panel?)

Relation to Prior Work: The authors discuss how their approach differs to previous efficient coding frameworks.

Reproducibility: Yes

Additional Feedback:


Review 2

Summary and Contributions: This paper derives a formal and general solution to the problem of optimal coding of a 1-dimensional variable, where optimality is defined with respect to a function of the Fisher information (and thus local discriminability) and subject to both explicit resource constraints and implicit constraints of monotone continuity of the encoding function. The result subsumes and generalizes a number of previous findings. The authors also use it to propose a scheme to determine the cost function and constraints

Strengths: While it builds on substantial earlier work in optimal coding theory, this paper offers a new and elegant, general formulation of the problem that is valuable in itself, and may well seed further progress in characterizing neural tuning and adaptation.

Weaknesses: The paper shares some limitations with much previous work (some of which are mentioned in the Discussion): a limitation to 1-D stimuli, a treatment of coding in isolation from computation, and a combination of a local discriminability measure with smooth monotone encoding (rather than a consideration of more general encoding schemes under mutual information). It is unfortunate that there is no direct link to known neural codes, nor a discussion of how known adaptation results may be interpreted in light of these arguments.

Correctness: I did not notice any errors.

Clarity: Yes.

Relation to Prior Work: Yes.

Reproducibility: Yes

Additional Feedback: I have noted the author feedback, but see no reason to revise my review.


Review 3

Summary and Contributions: The submission “Adaptation properties allow identification of optimized neural codes” describes a new formulation of efficient coding theory. This work builds upon several pieces of previous work. It uses a slightly more flexible efficient coding formulation than some of the previously work, by having an objective function that is based on the Fisher information while also considering the constraints on the neural activity. This theory does not directly operate on the firing rate, rather it operates on a more abstract quantify Fisher information. I think the model formulation is quite clever. The authors derived a set of results by consider different instantiations of the model. While in some cases it recovers known results in the literature, there are a few instances which have not been considered previously. The authors also consider the problem of recovering the objective function and the constraints from the neural data.

Strengths: Although I didn’t check every single line of math, the theoretical deviations in the paper appear to be sound for the most part. The theoretical formulation is general, yet could be used to make specific predications. The theory generalizes and helps organize some the recent results in efficient coding in a coherent framework. Efficient coding is an important topic in neural coding, and is clearly relevant to the Neuroscience community at NeurIPS. While one might consider the paper to be incremental, I do find the general formation presented in the paper to be quite interesting and informative. The paper also considers how to reverse-engineer the cost function and constraint under certain conditions. This is a previously overlooked problem in my view. Although the assumptions may seem to be a bit restrictive, these kind of exercises are still insightful.

Weaknesses: The writing of the manuscript is quite dense, and occasionally it is difficult to follow the arguments and understand the key points the authors would like to deliver. The paper would be stronger if the authors could include real dataset from physiology。

Correctness: I found the overall claims and method to be sound.

Clarity: Section 2.3 asks the question of “how efficient codes depend on neural noise”. However, I didn’t find a clear answer there. Maybe the authors could re-write to make it clear. Line 235-236, it is confusing to talk about p(\theta) “starts” to depend on p(s), because here the model didn’t consider time explicitly. Line 255-262, this is an important aspect of the results. However, the logic here is bit difficult to follow. Maybe it would be useful to first describe the key ideas and intuitions, then followed by the details.

Relation to Prior Work: Overall, the relation to the previous work is well described. The only thing that is unclear to me is how the results on deviation from activity constraints relate to the results in ref [14].

Reproducibility: Yes

Additional Feedback: A major technical concern I have is that how the authors can still recover the objective function and constraints while estimating the neural activity parameters. From the description in Sec. 2.4.3, in order to recover the objective and constraint functions, the code needs to adapt quickly. If that’s the case, how to reliably estimate the neural parameters in the presence of a change of the code? I’d thought that the two things would be disentangled. Added after the rebuttal period: After reading through other reviews and rebuttal, I remain positive about this paper. As other reviewers also pointed out, although the results in the paper are a bit incremental, but this (slightly) general framework formulated in the paper might still count as an interesting contribution. writing- The paper is quite dense, and the authors went over too many things- I hope the authors could further improve the presentations. One weakness is there are no experimental data presented, but if viewing as a theoretical paper, one could still justify it as a solid paper.


Review 4

Summary and Contributions: 1. This paper proposes an efficient coding formulation with adaptation to changing world with the following three ingredients in a 1D parameter (the sufficient statistics of stimulus) and a 1D stimulus space: * objective function (as a function of Fisher information) the code aims to optimize * constraints (like firing rate) on neural activities * stimulus distribution 2. The authors propose an experimental paradigm that can identify constraints on neural codes, regardless of the objective function, using a fixed-point iterative closed loop experiment, which might be used to evaluate animal's efficient coding constraints in certain adaptation settings.

Strengths: This is an interesting paper that links multiple important ideas in a tractable 1D model framework. It is appealing to test the brain's efficient coding strategy while the neural codes adapts a changing stimulus. This paper offers a simple yet clean way to generalize and formulate the brain's optimization algorithm by decomposing the adaptation part into the change of sufficient statistics p(theta|s) while keeping the encoding model p(r|theta) unchanged. An experiment is conceptually proposed to identify if a resource constraint exists experimentally. Such a prediction is of great importance for experimental validation of the theory.

Weaknesses: This is a hard paper to read. The authors should spend more time justifying and explaining some of their claims. I found the parts related to the flat Fisher space particularly obsure, which seems to be a critical aspect of the paper. The fixed point discussion could also be substantially improved. Within the scope of this paper, it's unclear if the optimization objective function is identifiable experimentally, although they do show that a deviation from a log is testable. Consequently, when the metabolic constraint is tested to be present (non-flat p(\hat(\theta))), it's hard to quantify the deviation since it will depend on the objective function form. No discussion for how the adaptation could be potentially implemented in the brain and predictable by this work.

Correctness: The derivations are clear. Basic assumptions are reasonable in the 1D case. Minor: [1] refers to Histogram Equalization, not Histogram Equilibration

Clarity: The section on information geometry needs more exposition and the flat Fisher space. It was unclear why the authors could move to the flat Fisher space by changing variables, and whether the reparameterization by \hat{\theta} was equivalent to altering the tuning of theta(s). L65: one-dimensional *manifold*, not subspace. Since theta is monotonically related to s, s will already be a sufficient statistic if theta is. Relatedly, this parameterization makes Fisher Info independent of noise, but why is the right one? \hat{\theta} not just a reparameterization of theta, but is also a reparameterization of s — and that was the original point of the optimization. **Update after rebuttal I thank the authors for their clarifications. I have decided to increase the score from 6 to 7.

Relation to Prior Work: Adequate. Mainly discussed in Section 1 and Section 2.1.

Reproducibility: Yes

Additional Feedback: It would be helpful to hold the reader's hand and provide more intuitions. These findings seem interesting but it is difficult for me to piece them together as is.

[Author Response · NeurIPS 2020]

We would like to thank all reviewers for their thoughtful and helpful feedback. In addition to incorporating helpful
edits, we have rewritten sections 2.3 and 2.4.3 and updated sections 2.4.2 and 2.5 to clarify their key points. Below we
highlight the essence of these changes.

**Parameterizations and parametric assumptions (R1,R3,R4)**

(**R1**) The model that we investigate splits into two pieces: an encoding model, $p(\boldsymbol{r}|\theta)$, and an optimized mapping from
stimuli to parameters, $\theta(s)$. This split allows us to study features of optimized codes that are agnostic to the encoding.
However, we can't connect these features to neural activity $\boldsymbol{r}$ without an encoding model. So, to apply our model to
neural data, we must either specify or fit such an encoding model. (**R1**, **R3**) In the example in section 2.5, the encoding
model is given by $p(\boldsymbol{r}|\boldsymbol{\eta}(\theta))$ where $p(\boldsymbol{r}|\boldsymbol{\eta})$ is an assumed exponential family distribution with parameters $\boldsymbol{\eta}$, and $\boldsymbol{\eta}(\theta)$
is a curve through the parameter space, which is fit, akin to generalized linear model regression. (**R3**) This allows us to
separate the question of parameter fitting (i.e., finding $\boldsymbol{\eta}(\theta)$) from the question of adaptation (i.e., identifying how $\theta(s)$
changes with $p(s)$). The parameter space curve, $\boldsymbol{\eta}(\theta)$, can be fit to data from any stimulus distribution. This curve,
$\boldsymbol{\eta}(\theta)$ is assumed to be fixed within the timescale of an experiment, while adaptation to different stimulus distributions,
$p(s)$, determines $\theta(s)$. The timescale difference is essential for our model to be applicable, and can be checked in data
by fitting the parameter curves $\boldsymbol{\eta}(\theta(s))$ for different stimulus distributions to ensure that they differ only in $\theta(s)$. This
discussion has been added to sections 2.4.3 and 2.5.

(**R4**) The parameters, $\theta$, flat Fisher space parameters, $\hat{\theta}$, and stimuli, $s$, are all sufficient statistics of $s$ in the neural
activity. So, what do we gain from changing $s \to \theta$ to $s \to \hat{\theta}$, and what is the role of the latter? The parameter $\theta$ is
inherited from our parameterization of the encoding model $p(\boldsymbol{r}|\theta)$, but is not unique. In fact, we could choose post-hoc
a new parameterization $\omega(\theta)$ so that the new parameters have any optimal distribution, $p_\omega(\omega)$, of our choice, with
associated Fisher information $\sqrt{I_\omega} = \sqrt{I_\theta}\, d\theta/d\omega$. The flat-Fisher parameter $\hat{\theta}$ is the unique choice of $\omega$ for which
this Fisher information becomes constant, or 'flat'. Furthermore, $p_{\hat{\theta}}$ is fully determined from measurable quantities,
$p_{\hat{\theta}} = p_s/\sqrt{I_s}$. Thus the flat Fisher condition renders $p_{\hat{\theta}}$ independent of our initially chosen (potentially arbitrary)
parameterization, and therefore reflects features of the code irrespective of this choice. (**R4**, **R1**) This highlights an
important insight gained from working with general encoding models: the flat space distribution $p_{\hat{\theta}}$ reflects intrinsic
features of the code adaptation, while other parameterizations can be confounded by features of their assumed encoding
models. Previous approaches, e.g., Wei & Stocker (2015), have worked in a similar space, arguing from examples.
Section 2.3 has been rewritten and updated to reflect this intuition.

**Fixed point iteration approach (R1,R2,R3,R4)**

(**R1**) The fixed point of neural adaptation has two key features. First, it depends only on the constraint, and has the
form: $p_{\hat{\theta}} \propto \exp(\lambda C(\theta(\hat{\theta})))$. This closed-form expression for the fixed point guarantees that a fixed point will exist.
Second, this fixed point can be experimentally identified by a *closed-loop experimental scheme*, where the neural
activity measured in response to stimuli drawn from one stimulus distribution determines the stimulus distribution that
will be presented next. Briefly, we allow the neural code to adapt to an initially uniform stimulus distribution, $p_s(s) \propto 1$,
and then measure neural responses $\boldsymbol{r}_i$ to stimuli $s_i \sim p_s$. From these responses, we fit the flat space distribution,
$p_{\hat{\theta}}$, which then becomes the new stimulus distribution. The process of re-adaptation, measurement, and updating the
stimulus distribution repeats until there are no changes to the stimulus distribution. At this point, $p_{\hat{\theta}} = p_s$, and the
adaptation fixed point is achieved. The fact that this fixed point distribution only depends on the constraint function,
$C(\theta(\hat{\theta}))$, allow us to fit the constraint. (**R4**) Once we have recovered the constraint function from the last step of the
iteration, we can fit the objective function from the first iteration step. For this step, the stimulus distribution is uniform,
so the flat space parameter distribution has the form $p_{\hat{\theta}} \propto 1/\hat{f}^{-1}(\lambda C(\theta(\hat{\theta}) + Z)$, allowing the objective function to be
fit using the constraint function that we already know. We have rewritten section 2.4.3 to clarify these points.

(**R2**, **R3**) Along with improving methods for fitting encoding models, application of these methods to neural data
remains an high priority. The fixed-point procedure we described is a closed-loop experimental paradigm, and so is
difficult to apply outside of this context. This doesn't preclude application of other parts of our theory to existing
experimental data. Fairhall et al. (2001) and Brenner et al. (2000), for example, observed that the distribution of activity
parameters (firing rate) was not impacted by (rapid) re-adaptation to different stimulus distributions. According to our
log objective function test this implies a log Fisher information objective for the measured codes. This is reasonable
for early sensory neurons, since the log objective function approximates mutual information. This was added as an
example to section 2.4.2.

**Revised statement of broader impact (R1,R3)**

(**R1**, **R3**) We have revised the statement of broader impact to read: "This work builds toward improved understanding
of the neural code, and thus, a better understanding of operation of the nervous systems and the behavior of humans and
other animals. Such understanding is important for neural prostheses and may lead to novel treatment options of human
brain diseases. One potential risk is that a better understanding of neural coding, particularly value coding, lends itself
to abuse for human behavioral modification. This is not an immediate concern for the work presented here."

[Meta-Review · NeurIPS 2020]

Four knowledgeable referees support acceptance for the contributions, notably for proposing a creative new variant on past studies of efficient coding, and I also recommend acceptance. The reviewers were happy with the rebuttal which helped clarify a number of ambiguities and two reviewers increased their score following the rebuttal/discussion.